# MCNC: Manifold-Constrained Reparameterization for Neural Compression

**Chayne Thrash**[1], **Ali Abbasi**[1], **Reed Andreas**[1], **Parsa Nooralinejad**[2],
**Soroush Abbasi Koohpayegani**[2], **Hamed Pirsiavash**[2], **Soheil Kolouri**[1]

[1]Department of Computer Science, Vanderbilt University, Nashville, TN, 37235
[2]Department of Computer Science, University of California, Davis, CA, 95616

`chayne.thrash,ali.abbasi,reed.w.andreas,soheil.kolouri@vanderbilt.edu`
`pnoorali,soroush,hpirsiav@ucdavis.edu`

## Abstract

The outstanding performance of large foundational models across diverse tasks, from computer vision to speech and natural language processing, has significantly increased their demand. However, storing and transmitting these models poses significant challenges due to their massive size (e.g., 750GB for Llama 3.1 405B). Recent literature has focused on compressing the original weights or reducing the number of parameters required for fine-tuning these models. These compression methods generally constrain the parameter space, for example, through low-rank reparametrization (e.g., LoRA), pruning, or quantization (e.g., QLoRA) during or after the model training. In this paper, we present a novel model compression method, which we term Manifold-Constrained Neural Compression (MCNC). This method constrains the parameter space to low-dimensional pre-defined and frozen nonlinear manifolds, which effectively cover this space. Given the prevalence of good solutions in over-parameterized deep neural networks, we show that by constraining the parameter space to our proposed manifold, we can identify high-quality solutions while achieving unprecedented compression rates across a wide variety of tasks and architectures. Through extensive experiments in computer vision and natural language processing tasks, we demonstrate that our method significantly outperforms state-of-the-art baselines in terms of compression, accuracy, and/or model reconstruction time. Our code is publicly available at https://github.com/mint-vu/MCNC.

## 1 Introduction

Recent state-of-the-art performance in computer vision (Dosovitskiy et al., 2021; Zhai et al., 2022), natural language processing (Brown et al., 2020; Touvron et al., 2023), speech (Zhang et al., 2023b; Baevski et al., 2020), and other fields (Grechishnikova, 2021; Jumper et al., 2021) is often achieved through large transformer networks (Vaswani et al., 2017) with hundreds of millions to tens of billions of parameters, trained on vast amounts of data. For example, the Llama 3.1 model (Dubey et al., 2024) with its 405B parameters requires around 750GB of memory. With the growing size of models and the shift towards large foundational models, effective compression methods are increasingly vital to reduce size without significant performance loss, enabling efficient storage, communication, and deployment on edge devices.

Fine-tuning foundation models for custom tasks and datasets is now the gold standard for producing high-performing, task-specific models. However, this approach raises concerns about the storage of these customized models and creates a computational bottleneck when loading task-specific models and transferring their weights from CPU to GPU. To address these issues, Parameter Efficient Fine-Tuning (PEFT) methods have gained popularity in recent years. These methods focus on efficiently compressing the residual weights or parameters needed to adapt foundation models to specific tasks. For example, LoRA (Hu et al., 2022) imposes low-rank constraints on residual weights, reducing the number of trainable parameters required for fine-tuning, making it a widely adopted approach.

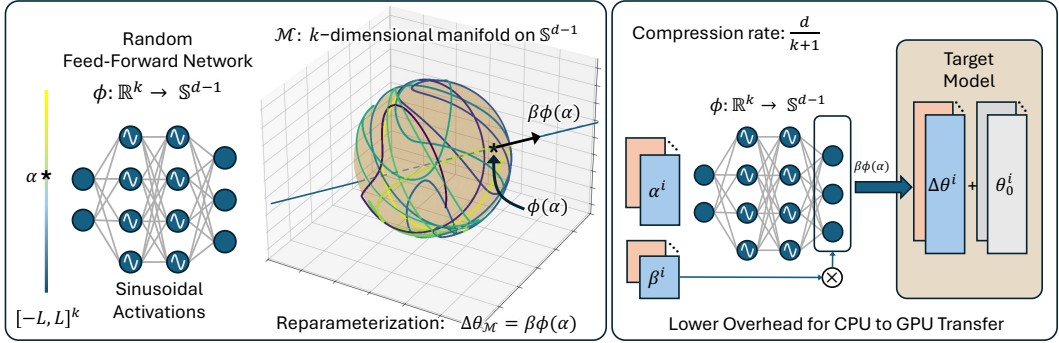

Figure 1: Illustration of our proposed reparameterization technique, Manifold Constrained Network Compression (MCNC). The parameters $\theta \in \mathbb{R}^d$ are decomposed as $\theta = \theta_0 + \Delta\theta$, where $\theta_0$ is fixed and $\Delta\theta = \beta u$ represents the learnable residuals. Here, $\beta$ is the amplitude, and $u \in \mathbb{S}^{d-1}$ is a unit vector on the $d$-dimensional hypersphere. The unit vector $u$ is generated by mapping a lower-dimensional vector $\alpha \in \mathbb{R}^k$ through a nonlinear generator $\phi : \mathbb{R}^k \to \mathbb{S}^{d-1}$, allowing the learnable perturbation to lie within a $k$-dimensional subspace wrapped around the hypersphere. The right panel depicts the model partitioning strategy, where the weights are divided into $d$-dimensional chunks, and the corresponding $(\alpha, \beta)$ pairs are learned for each chunk.

A wide variety of methods have been proposed to compress large models (Tang et al., 2024). These approaches can generally be grouped into five main techniques: weight sharing, quantization, pruning, knowledge distillation, and reparameterization. Relevant to our method, the reparameterization approaches (Hu et al., 2022; Nooralinejad et al., 2023; Girish et al., 2023; Koohpayegani et al., 2024) aim to reduce the number of parameters or simplify the computations by restructuring the model weights. This often involves parameterizing the weights through more efficient forms, such as low-rank matrices (Hu et al., 2022), which can maintain the model's expressiveness while significantly reducing computational and memory overhead. In this paper, we introduce a novel non-linear reparameterization technique for model compression, which achieves unprecedented performance at extreme compression rates while reducing the CPU-to-GPU transfer time when loading networks.

Our work in this paper draws inspiration from recent reparameterization methods such as PRANC (Nooralinejad et al., 2023) and NOLA (Koohpayegani et al., 2024). These methods take advantage of the fact that overparameterized neural networks with $d$ parameters often have a large number of good solutions (Liu et al., 2022). Hence, by restricting the search for optimal solutions to random $k$-dimensional subspaces within the parameter space, where $k \ll d$, they provide significant compression, i.e., $\frac{d}{k}$, without sacrificing performance. In this paper, we take a different approach and ask: Could better solutions be found by expanding the complexity of the search space from a random subspace (Nooralinejad et al., 2023; Koohpayegani et al., 2024) to a $k$-dimensional nonlinear manifold that more efficiently captures the structure of the parameter space? Below, we introduce the core idea of our method through a thought experiment.

**Winding a string around a sphere.** Consider optimizing a model with parameters $\theta \in \mathbb{R}^d$, where $\theta = \theta_0 + \Delta\theta$, with $\theta_0$ fixed. Using polar decomposition, we express $\Delta\theta = \beta u$, where $\beta \in \mathbb{R}$ is the amplitude and $u = \frac{\Delta\theta}{\|\Delta\theta\|_2} \in \mathbb{S}^{d-1}$ represents the direction on the hypersphere $\mathbb{S}^{d-1}$. Now, consider a segment of the real line $[-L, L]$, representing a string of length $2L$, wrapped around $\mathbb{S}^{d-1}$, parameterizing a one-dimensional manifold with $\alpha \in [-L, L]$. Instead of optimizing in $d$-dimensional space, we optimize over the amplitude $\beta$ and the manifold parameter $\alpha$. Figure 1 (left panel) illustrates this for $d = 3$. Extending this, we replace the segment with a $k$-dimensional space $[-L, L]^k$, wrapping it around $\mathbb{S}^{d-1}$ to increase coverage. This reduces the parameter space from $d$ to $k + 1$, reparameterizing $\theta$ by $\alpha$ and $\beta$. To wind such a $k$-dimensional subspace around the $d$-dimensional hypersphere $\mathbb{S}^{d-1}$, we propose to use a random feedforward neural network with sinusoidal activations, which we refer to as the 'random generator.' Sinusoidal activations are essential because they introduce periodicity in parameterization, facilitating smoother and more uniform coverage of the hypersphere and enhancing the differentiability of the generator.

**Contributions.** Our specific contributions in this work are:

1. Introducing a novel non-linear reparameterization technique for model compression, which we term Manifold-Constrained Neural Compression (MCNC). This method restricts the

optimization of network parameters to a $k$-dimensional manifold within the original $d$-dimensional parameter space, enabling more efficient compression.

2. Demonstrating the effectiveness of our proposed method compared to recent network compression methods in the literature across vision and natural language processing tasks and across diverse architectures.

## 2 RELATED WORK

Our goal in this paper is to train a deep model with a minimal number of parameters, making it more efficient to communicate models between agents/entities or store on devices with limited memory. Our compact representation makes no assumptions about the model size, weight distribution, the number of non-zero parameters, or computational precision, making it orthogonal to methods like weight-sharing, quantization, pruning, and knowledge distillation. Hybrid approaches could integrate Manifold Constrained Neural Compression (MCNC) with these techniques. In the following, we briefly review these methods, which aim to reduce the model size and improve the inference time.

**Weight sharing** can be enforced through the model architecture, such as with convolutional kernels (Krizhevsky et al., 2012), recurrent networks (Hochreiter & Schmidhuber, 1997), or other hybrid architectures (Dehghani et al., 2019). It can also be achieved post-training via clustering mechanisms (Nowlan & Hinton, 1992; Ullrich et al., 2017) or hashing (Chen et al., 2015; Eban et al., 2020; Reagan et al., 2018) to bundle the weights of a model into a few shared components. Other notable weight sharing methods proposed in recent years include (Gao et al., 2019; Plummer et al., 2022; Shakerinava et al., 2024). Similar to weight sharing methods, MCNC uses a trains weights in a higher dimensional space using a low dimensional representation. However, MCNC is a general reparameterization that can be combined with weight-sharing techniques, e.g., layers in the network can share low-dimensional parameters to enhance compression.

**Quantization** is a widely used model compression technique that reduces the bit-width needed to represent model weights and activations. For instance, Rastegari et al. (2016) proposed XNOR-Networks, where binary operations are used in a network with XNOR gates to approximate convolutions, leading to a reported $58\times$ speedup and $32\times$ memory savings. The core challenge with these techniques is that naive quantization of models often results in a significant performance drop. On the other hand, quantized training/fine-tuning is often formulated as a constrained optimization problem that requires specialized optimization techniques (Lee et al., 2021). A large body of recent work focuses on effective quantization methods for large models (Xiao et al., 2023; shkolnik et al., 2020; Yao et al., 2023; Dettmers et al., 2024). In the extreme case, quantization reduces float32 to binary, achieving $32\times$ compression. In this paper, we target higher compression rates.

**Pruning** is a primary technique for network compression, as highlighted in several studies (Hassibi et al., 1993; Kim et al., 2020; Lin et al., 2020; Siems et al., 2021; Tiwari et al., 2021; Hayou et al., 2020; Wang et al., 2020a; Li et al., 2021; Rastegari et al., 2016; Lee et al., 2021). High compression rates are achievable through methods like those in (Kusupati et al., 2020b; Isik et al., 2022; Zhang et al., 2022; Sun et al., 2024), making pruning one of the main methods for network compression. Unstructured pruning zeros out many parameters, storing only the non-zero weights and their indices. Compression efficiency could further be improved with coding strategies like Huffman or Run-Length Encoding (RLE) (Han et al., 2015a). More recently, various works have considered structured/structural pruning removing entire modules from a network while preserving the performance (Anwar et al., 2017; Fang et al., 2023). We show that MCNC matches the performance of pruning methods at lower compression rates while outperforming them at higher rates.

**Compression by reparameterization** has recently emerged as a powerful method for efficiently compressing large models. LoRA (Hu et al., 2022) introduced low-rank reparameterization for fine-tuning large models, inspiring many extensions (Valipour et al., 2023; Zhang et al., 2023a; Dettmers et al., 2024; Koohpayegani et al., 2024). PRANC (Nooralinejad et al., 2023), closely related to our work, constraints model weights to a randomly selected low-dimensional subspace. NOLA (Koohpayegani et al., 2024) combines ideas from PRANC and LoRA, pushing low-rank PEFT below the classic rank-one limit. NOLA and PRANC assume that good solutions lie on low-dimensional nonlinear manifolds within the overparameterized model's parameter space (Liu et al., 2022; Li et al., 2018), making it likely that a random subspace will intersect this manifold, allowing for viable solutions within the subspace.

This paper stems from our curiosity to explore whether a random $k$-dimensional manifold in a $d$-dimensional parameter space ($k \ll d$) can be constructed to maximize coverage, thereby increasing the likelihood of intersecting the manifold of good solutions. Hence, our method can be considered as a generalization of PRANC (Nooralinejad et al., 2023) and NOLA (Koohpayegani et al., 2024).

**Manifold constrained optimization** has been employed in various works to constrain model parameters to a Riemannian manifold in order to induce various geometric characteristics (Fei et al., 2023). For instance, some methods require that the model parameters reside on or be close to the Stiefel manifold, i.e., the set of $k$ orthonormal vectors in $\mathbb{R}^d$, to reduce redundant representations (Ozay & Okatani, 2016; Huang et al., 2018; Wang et al., 2020b). Manifold-constrained optimization (Absil et al., 2008; Boumal, 2023) in neural networks is gaining significant attention, as evidenced by recent developments in geometric optimization libraries for deep learning (Meghwanshi et al., 2018; Kochurov et al., 2020; Luchnikov et al., 2021). In this work, we use a random feed-forward network with sinusoidal activations to parameterize a low-dimensional manifold within a higher-dimensional hypersphere. We constrain optimization to this manifold, relying exclusively on standard (stochastic) gradient descent, avoiding the need for Riemannian optimization.

## 3 METHOD

At its core, Manifold Constrained Neural Compression (MCNC) relies on two key components: 1) an explicit nonlinear mapping that wraps a $k$-dimensional space around a sphere in a $d$-dimensional space, where $k \ll d$, and 2) partitioning the model's parameters into $d$-dimensional segments and optimizing them in the $k$-dimensional input space of the nonlinear map. This process would result in a compression rate of almost $d/k$.

**Notations and Preliminaries.** Let $\theta \in \mathbb{R}^d$ denote a partition of the model parameters, which we aim to optimize. Through polar decomposition, we express $\theta$ by its amplitude $\beta = \|\theta\|$ and its direction $u = \theta/\|\theta\|_2$. Additionally, we denote the $d$-dimensional unit sphere with $\mathbb{S}^{d-1}$. Our goal is to reparameterize $u \in \mathbb{S}^{d-1}$ using significantly fewer parameters than $d$. To achieve this, we will employ a 'generator' model to explicitly parameterize a $k$-dimensional manifold that best represents the $d$-dimensional hypersphere.

### 3.1 TRAVERSING A HIGH-DIMENSIONAL SPHERE BY A LOW-DIMENSIONAL MANIFOLD

**Rationale.** We aim to model a $d$-dimensional hypersphere using a $k$-dimensional manifold. For clarity, let's examine this concept in a simpler scenario: consider $d = 3$, corresponding to a sphere, and $k = 1$, akin to a string. This familiar setting helps illustrate our objective. Given a string of a specific length, how can we most effectively cover the surface of the sphere? The answer is intuitive: simply wrap the string around the sphere. This wrapping acts as a nonlinear operator that takes a straight line and deforms it around the sphere. Similarly, a $k$-dimensional input space can be wrapped via a nonlinear function, i.e., a generator, around a hypersphere in the $d$-dimension. Next, we discuss how to formalize this problem.

**Formalizing the problem.** To traverse a hypersphere using a low-dimensional manifold, we aim to maximize its coverage. We formalize this problem as follows: Let $\mathcal{U}([-L, L]^k)$ be the uniform distribution over the $k$-dimensional hypercube $[-L, L]^k$, and $\mathcal{U}(\mathbb{S}^{d-1})$ the uniform distribution over the $d$-dimensional hypersphere $\mathbb{S}^{d-1}$. Our goal is to develop a nonlinear mapping that wraps the $k$-dimensional hypercube around the $d$-dimensional hypersphere, maximizing coverage. This problem can be formalized as finding a nonlinear function $\phi : \mathbb{R}^k \to \mathbb{R}^d$ that transforms $\mathcal{U}([-L, L]^k)$ into $\mathcal{U}(\mathbb{S}^{d-1})$, mapping samples $\alpha \sim \mathcal{U}([-L, L]^k)$ to the hypersphere such that $\phi(\alpha) \sim \mathcal{U}(\mathbb{S}^{d-1})$. Hence, we need to have a way to measure the uniformity of the $\phi(\alpha)$s. To do that, we measure the Wasserstein distance (Peyré et al., 2019) between the output probability distribution of $\phi$ and the uniform distribution on the hypersphere.

**Modeling the generator.** We model the generator, $\phi : \mathbb{R}^k \to \mathbb{R}^d$, via a feed-forward network. We consider various activation functions, including the Sigmoid, Rectified Linear Unit (ReLU), and Sine activation (Sitzmann et al., 2020). First, we ask whether a randomly initialized network can provide the desired characteristics. Second, we ask whether we can optimize such a generator to provide maximal space traversal. We note that optimizing such generator model mirrors the challenges of deep generative modeling, a subject thoroughly investigated in literature (Goodfellow et al., 2014; Arjovsky et al., 2017; Deshpande et al., 2018). We emphasize that our decision to use the uniform distribution on $\mathbb{S}^{d-1}$ as our target distribution stems from an assumption of no prior knowledge about

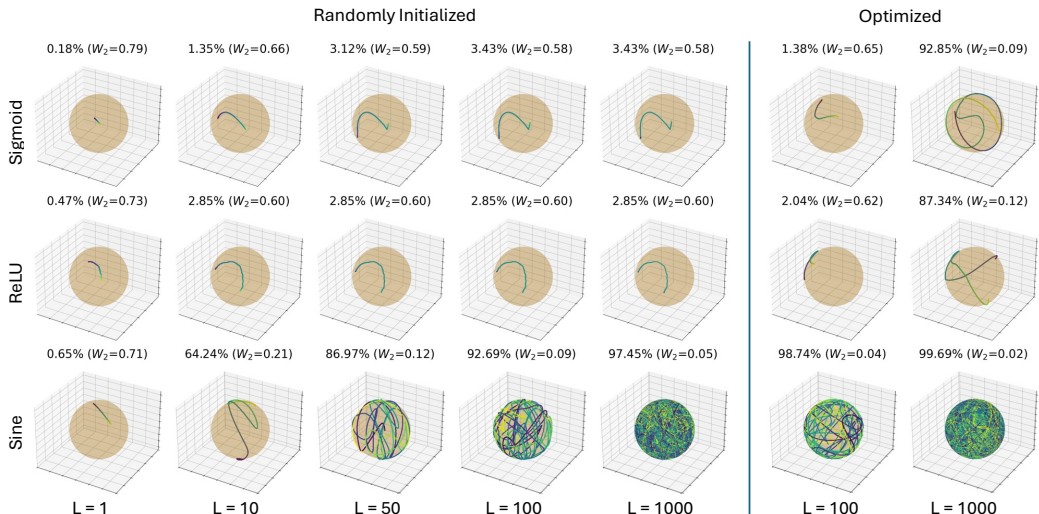

Figure 2: Traversal of a sphere using a 1-dimensional manifold, where $\phi : \mathbb{R} \to \mathbb{S}^2$ is a multi-layer perceptron with architecture $1 \to 1024 \to 1024 \to 3$ and activation functions Sigmoid, ReLU, or Sine. The input bound $L$ is absorbed into the first layer's weights. The left panel displays outputs of randomly initialized networks for different activations and $L$ values, while the right panel shows outputs after optimization. Uniformity is quantified using $\exp(-\tau W_2^2(\hat{\mu}, \nu))$, with $\tau = 10.0$ and $W_2$ representing the Wasserstein distance between the network's output $\hat{\mu}$ and the uniform distribution $\nu$. the importance of different directions for the downstream task of optimizing a network. Should such information become available, the target distribution could be adjusted to reflect this knowledge, allowing for more precise wrapping around areas of greater importance. In this paper, we used the SWGAN (Deshpande et al., 2018) framework to train the generator due to its simplicity.

As a guiding example for developing our generator, we consider the case where $k = 1$, $d = 3$, and $\phi$ is modeled as a feed-forward network with the architecture $1 \to 1024 \to 1024 \to 3$. Figure 2 shows the output of $\phi$ for randomly initialized networks with various activation functions, along with the outputs after optimization. We evaluate uniformity by reporting $\exp(-\tau W_2^2(\hat{\mu}, \nu))$, where $\hat{\mu}$ is the output distribution of $\phi$ and $\nu$ represents the uniform distribution on $\mathbb{S}^{d-1}$. These results are presented in Figure 2. Interestingly, we observe that for larger values of $L$, the randomly initialized network with Sine activations achieves strong coverage of the sphere, with optimization only marginally improving the coverage. Therefore, in our main results, we use a randomly initialized feed-forward network with Sine activations as the generator while also conducting an ablation study on the impact of training the generator. Notably, the random generator can be efficiently stored or communicated using a scalar random seed, assuming access to a shared pseudo-random number generator (PRNG).

## 3.2 REPARAMETERIZATION AND MANIFOLD CONSTRAINED OPTIMIZATION

Given a random generator model, $\phi : \mathbb{R}^k \to \mathbb{S}^{d-1}$, we reparameterize a $d$-dimensional residual vector as $\Delta\theta = \beta\phi(\alpha)$, where $\beta \in \mathbb{R}$ and $\alpha \in \mathbb{R}^k$, thereby reducing the number of parameters from $d$ to $k + 1$. Figure 1 illustrates this concept. Let $\mathcal{L} : \mathbb{R}^d \to \mathbb{R}$ denote the loss function for a specific task of interest, e.g., image classification, where $\mathcal{L}(\theta)$ is the associated loss with parameter $\theta$. We then constrain the training of the parameter $\theta$ as:

$$\theta^* = \theta_0 + \beta^*\phi(\alpha^*), \quad (\alpha^*, \beta^*) = \underset{\alpha, \beta}{\arg\min}\, \mathcal{L}(\theta_0 + \beta\phi(\alpha)) \tag{1}$$

Note that we use $\theta_0$ to emphasize that optimization can begin from any random initialization or pre-trained weights, such as those used in PEFT. This optimization process constrains the model parameters, $\theta$, to lie on a $k$-dimensional manifold within $\mathbb{S}^{d-1}$, which is parameterized by $\phi$.

## 3.3 LEARNING THE DEEP MODEL

We initialize the generator $\phi(.)$ randomly once and freeze it. Then, given a deep model, we reshape its parameters to a long vector and then divide that into chunks of size $d$, and reparameterize each chunk with $k + 1$ parameters using the generator $\phi(.)$. In the case the model size is not divisible by

Table 1: **Comparison of our method with pruning techniques on ViT-Ti and ViT-S models trained on ImageNet-100.** We exclude position embeddings, CLS token, and Layer Normalization parameters when computing the percentage of model size.

| ViT-Ti | | |
|---|---|---|
| **Method** | **Percentage of Model Size** | **Acc.** |
| Baseline | 100% | 83.5 |
| Magnitude 2015b (Pr. 66.7%) | | 80.4 |
| PLATON 2022 (Pr. 66.7%) | 50% | **81.6** |
| MCNC (Ours) | | 80.0 |
| Magnitude 2015b (Pr. 86.7%) | | 73.0 |
| PLATON 2022 (Pr. 86.7%) | 20% | 76.1 |
| MCNC (Ours) | | **77.1** |
| Magnitude 2015b (Pr. 93.3%) | | 60.9 |
| PLATON 2022 (Pr. 93.3%) | 10% | 67.2 |
| MCNC (Ours) | | **72.9** |
| Magnitude 2015b (Pr. 96.7%) | | 45.8 |
| PLATON 2022 (Pr. 96.7%) | 5% | 55.0 |
| MCNC (Ours) | | **69.1** |
| Magnitude 2015b (Pr. 98.7%) | | 29.4 |
| PLATON 2022 (Pr. 98.7%) | 2% | 39.0 |
| MCNC (Ours) | | **61.3** |
| Magnitude 2015b (Pr. 99.3%) | | 21.2 |
| PLATON 2022 (Pr. 99.3%) | 1% | 30.6 |
| MCNC (Ours) | | **52.9** |

| ViT-S | | |
|---|---|---|
| **Method** | **Percentage of Model Size** | **Acc.** |
| Baseline | 100% | 83.9 |
| Magnitude 2015b (Pr. 83.3%) | | 79.8 |
| PLATON 2022 (Pr. 83.3%) | 25% | 82.2 |
| MCNC (Ours) | | **82.7** |
| Magnitude 2015b (Pr. 93.3%) | | 72.0 |
| PLATON 2022 (Pr. 93.3%) | 10% | 77.4 |
| MCNC (Ours) | | **80.2** |
| Magnitude 2015b (Pr. 96.7%) | | 66.0 |
| PLATON 2022 (Pr. 96.7%) | 5% | 71.1 |
| MCNC (Ours) | | **76.3** |
| Magnitude 2015b (Pr. 98.7%) | | 41.5 |
| PLATON 2022 (Pr. 98.7%) | 2% | 57.6 |
| MCNC (Ours) | | **66.7** |

$d$, the last chunk will have some extra parameters that will be ignored. Finally, we train the model by optimizing $(\alpha, \beta)$ for all chunks using Eq 1 to minimize the loss of the deep model on the task of interest, e.g., image classification. The backpropagation is as simple as calculating the gradient for model parameters and then using the chain rule to backpropagate through the generator $\phi(.)$. Hence, auto-differentiation can be directly used to optimize $\alpha$s and $\beta$ without the need for geometric optimization techniques.

## 4 EXPERIMENTS

To demonstrate the performance of our method, we evaluate MCNC under two different settings:

1. **Training from scratch for image classification.** In this setting, we let $\theta_0$ be a randomly initialized network and optimize using MCNC. Note that since a randomly initialized network can be communicated using only the random seed, this does not increase the cost of compression. We show our effectiveness at compressing both Vision Transformer (ViT) (Dosovitskiy et al., 2021) and ResNet (He et al., 2016) architectures.

2. **Parameter Efficient Fine-Tuning of LLMs.** In this setting, we instead begin with a pre-trained $\theta^*$ and optimize the $\Delta\theta$ via MCNC. We conduct this experiment for LLMs where fine-tuning of large models has become the norm.

Given that our method is orthogonal to the low-rank parameterization used in LoRA (Hu et al., 2022), we reparameterize either the original networks or their rank-constrained versions in our experiments.

### 4.1 TRAINING IMAGE CLASSIFIERS FROM SCRATCH

**ImageNet-100 on Vision Transformer Architectures.** We begin by evaluating on training Vision Transformers from scratch on the ImageNet-100 dataset (Tian et al., 2020). We evaluate on both ViT-Ti and ViT-S (Touvron et al., 2021). We compare MCNC against iterative unstructured pruning method PLATON (Zhang et al., 2022) as well as the classic Magnitude pruning (Han et al., 2015b). We report the final compressed model size as a percentage. As unstructured pruning also requires storing the indices of non-zero values, these must be accounted for when computing the compression rate. Although this requires storing two values per weight, the number of bits required for each index can be reduced by storing the distance between each non-zero value (Han et al., 2015a). Therefore, we assume half-precision for the indices and prune to sparsity rates 50% higher than the desired compression. We exclude position embeddings, CLS token, and Layer Normalization (Ba et al., 2016) parameters from all compression methods. We provide hyperparameters used for our method and baselines in section A.3. Table 1 show our results on ViT-Ti and ViT-S. Our results show that, particularly for high compression rates, MCNC is more effective than pruning for compression.

Table 2: **Comparison with PRANC and NOLA on ImageNet-100 using ResNet-18** with multiple compression rates.

| Method | Ref | Percentage of Model Size | Acc. |
|---|---|---|---|
| Baseline | - | 100% | 82.1 |
| Ours | - | 50% | $80.7 \pm 0.4$ |
| Ours w/ LoRA | - | | $79.1 \pm 0.7$ |
| Ours | - | 20% | $79.9 \pm 0.3$ |
| Ours w/ LoRA | - | | $80.0 \pm 0.2$ |
| Ours | - | 10% | $78.0 \pm 0.0$ |
| Ours w/ LoRA | - | | $80.3 \pm 0.5$ |
| Ours | - | 5% | $75.4 \pm 0.2$ |
| Ours w/ LoRA | - | | $79.6 \pm 0.1$ |
| PRANC 2023 | 2023 | 2% | 67.3 |
| Ours | - | | $70.2 \pm 0.1$ |
| Ours w/ LoRA | - | | $\mathbf{76.9 \pm 0.2}$ |
| PRANC 2023 | 2023 | 1% | 61.1 |
| NOLA 2024 | 2024 | | 64.7 |
| Ours | - | | $63.4 \pm 0.2$ |
| Ours w/ LoRA | - | | $\mathbf{71.7 \pm 0.3}$ |

For ViT-Ti, we outperform pruning by $8\%$ at $10\%$ of the original model size. On ViT-S, although pruning methods are able to outperform at higher compression rates, we see that our method is able to maintain high performance as we decrease model size.

**ImageNet-100 on ResNet-18.** Next, we show that our method works across architectures by training from scratch on ImageNet-100 using the ResNet-18 (He et al., 2016) architecture. We compare with other recent network reparameterization methods PRANC (Nooralinejad et al., 2023) and NOLA (Koohpayegani et al., 2024). Since it was shown in (Koohpayegani et al., 2024) that reparameterizing LoRA rather than the original model architecture can boost performance, we also show results of our method in this setting. Results are shown in Table 2. Our method demonstrates a $7\%$ accuracy improvement over the baselines at $1\%$ of the original model size. In addition, as we reduce the compression rate we see that we can recover up to $97\%$ of the original model accuracy at $10\%$ of the size.

**CIFAR-10 and CIFAR-100.** We compare MCNC on CIFAR-10 and CIFAR-100 (Krizhevsky et al., 2009) with both pruning methods as well as PRANC (Nooralinejad et al., 2023) and NOLA (Koohpayegani et al., 2024) under extreme compression rates. We show results on the two datasets using both ResNet-20 and ResNet-56. For the pruning methods and PRANC, we directly compare with the results reported in (Nooralinejad et al., 2023). We present results for these experiments in Table 3. It can be seen that MCNC is able to consistently outperform all other methods while using a similar number of parameters. In addition, just as NOLA improves upon PRANC with the addition of LoRA, MCNC gains a boost as well.

## 4.2 MCNC FOR FINE-TUNING LARGE LANGUAGE MODELS

In this section, we compared the performance of MCNC and NOLA in parameter-efficient fine-tuning. We conducted fine-tuning experiments on two variants of the LLaMA-2 (Touvron et al., 2023) language model, LLaMA 7B, and LLaMA 13B, using the Alpaca dataset as our training data (Taori et al., 2023). We report both training loss and validation loss on the Alpaca dataset. Additionally, we report MMLU (Massively Multitask Language Understanding) (Hendrycks et al., 2021) 5-shot accuracy following the standard practice. This benchmark spans 57 tasks across diverse fields such as computer science, mathematics, law, and history.

**Implementation Details:** We use the training code from QLoRA (Dettmers et al., 2023) and NOLA (Koohpayegani et al., 2024) for our experiments. We quantize the original parameters of the language model to 4-bit and apply and fine-tune the adapter on all layers of the transformer. For NOLA, we followed the hyperparameters reported in (Koohpayegani et al., 2024). We set the rank to $8$ for both NOLA and our method in LLaMA 7B, and to $16$ in LLaMA 13B. In our method, we use a generator with the following specifications: a 3-layer MLP with an input dimension of 5, an output dimension of 5000, and a hidden dimension of 32. In NOLA, we use 64 bases for A and B in LLaMA 7B

Table 3: **Comparison of our method with pruning methods as well as PRANC and NOLA.** We include results for our method with and without LoRA applied to each of the layers. As none of the methods compress BatchNorm parameters, we remove them from all parameter counts.

| Method | Ref | Arch. | Dataset | # Params | Acc. |
|---|---|---|---|---|---|
| Baseline | - | R20 | C10 | $269,722$ | 88.9 |
| STR 2020a | 2023 | R20 | C10 | $12,238$ | 76.0 |
| PRANC 2023 | 2023 | R20 | C10 | $10,000$ | 81.5 |
| NOLA 2024 | 2024 | R20 | C10 | $11,500$ | 82.4 |
| MCNC (Ours) w/o LoRA | - | R20 | C10 | $10,380$ | $82.3 \pm 0.3$ |
| MCNC (Ours) w/ LoRA | - | R20 | C10 | $9,640$ | $\mathbf{83.9 \pm 0.1}$ |
| Baseline | - | R56 | C10 | $853,018$ | 91.6 |
| DPF 2020 | 2023 | R56 | C10 | $13,414$ | 47.7 |
| SuRP 2022 | 2023 | R56 | C10 | $10,834$ | 66.7 |
| STR 2020a | 2023 | R56 | C10 | $13,312$ | 67.8 |
| PRANC 2023 | 2024 | R56 | C10 | $5,000$ | 76.9 |
| NOLA 2024 | - | R56 | C10 | $5,000$ | $78.5 \pm 0.1$ |
| MCNC (Ours) w/o LoRA | - | R56 | C10 | $5,280$ | $78.7 \pm 0.2$ |
| MCNC (Ours) w/ LoRA | - | R56 | C10 | $5,590$ | $\mathbf{81.5 \pm 0.2}$ |
| Baseline | - | R20 | C100 | $275,572$ | 60.8 |
| DPF 2020 | 2023 | R20 | C100 | $10,770$ | 12.2 |
| SuRP 2022 | 2023 | R20 | C100 | $6,797$ | 14.5 |
| STR 2020a | 2023 | R20 | C100 | $10,673$ | 13.2 |
| PRANC 2023 | 2023 | R20 | C100 | $5,000$ | 32.3 |
| NOLA 2024 | - | R20 | C100 | $5,180$ | $35.6 \pm 0.1$ |
| MCNC (Ours) w/o LoRA | - | R20 | C100 | $5,110$ | $34.7 \pm 0.2$ |
| MCNC (Ours) w/ LoRA | - | R20 | C100 | $5,070$ | $\mathbf{36.7 \pm 0.2}$ |
| Baseline | - | R56 | C100 | $858,868$ | 64.3 |
| DPF 2020 | 2023 | R56 | C100 | $19,264$ | 19.1 |
| SuRP 2022 | 2023 | R56 | C100 | $10,919$ | 14.6 |
| STR 2020a | 2023 | R56 | C100 | $18,881$ | 26.0 |
| PRANC 2023 | 2023 | R56 | C100 | $5,000$ | 33.0 |
| NOLA 2024 | - | R56 | C100 | $5,000$ | $36.2 \pm 0.6$ |
| MCNC (Ours) w/o LoRA | - | R56 | C100 | $5,049$ | $35.2 \pm 0.1$ |
| MCNC (Ours) w/ LoRA | - | R56 | C100 | $5,015$ | $36.5 \pm 0.8$ |

Table 4: **Instruction finetuning for quantized LLaMA-2:** We fine-tune LLaMA-2 with the Alpaca dataset and compare MCNC to NOLA, maintaining the same number of parameters for both. "Generation GFLOPs" refers to the FLOPs needed to generate the original parameters of the adapter on the fly. MCNC achieves comparable performance to NOLA while requiring fewer FLOPs for on-the-fly parameter generation. This efficiency results in faster inference and training for MCNC compared to NOLA. We use a single RTX 3090 GPU to calculate the throughput. Note that the throughput here includes both the adapter's reconstruction and the forward pass through the base model and adapter.

| Method | Trainable Parameters | MMLU Acc | Train Loss | Val Loss | Throughput (Samples/Sec) | Adapter Model Reconstruction GFLOPs |
|---|---|---|---|---|---|---|
| LLaMA-2 - 7B (4-bit) | | | | | | |
| LoRA (rank=1) 2022 | 2.5M | 45.6 | 0.98 | 1.06 | 7.1 | - |
| NOLA 2024 | 28k | 45.5 | 1.08 | 1.01 | 3.1 | 2.56 |
| MCNC | 25k | 45.9 | 1.03 | 1.01 | 6.2 | 1.37 |
| LLaMA-2 - 13B (4-bit) | | | | | | |
| LoRA (rank=1) 2022 | 3.9M | 54.8 | 0.94 | 0.97 | 4.1 | - |
| NOLA 2024 | 78k | 54.8 | 1.00 | 0.96 | 1.7 | 17.53 |
| MCNC | 77k | 55.0 | 1.00 | 0.95 | 3.9 | 4.22 |

and 140 bases for LLaMA 13B. These numbers of bases result in the same number of optimized parameters as our method for each architecture. All other hyperparameters were identical for both methods except the learning rate; we used $lr = 0.001$ for NOLA and 0.01 for ours.

**Results:** The results are presented in Table 4. MCNC has comparable performance to NOLA with a similar number of parameters. However, MCNC requires $46\%$ fewer GFLOPs for generating

parameters on the fly compared to NOLA. This efficiency translates into faster inference and training for MCNC compared to NOLA. As shown in Table 4, MCNC achieves double the throughput for LLaMA 7B and 2.2 times higher throughput for LLaMA 13B. While it is feasible to generate parameters offline and merge them with the original weights of the language model, this approach limits the applications when processing multiple tasks and their corresponding adapters in a batch. Therefore, in scenarios involving batch processing of tasks, MCNC holds an advantage over NOLA due to its faster throughput.

## 4.3 ABLATION STUDIES

In the previous experiments, we've shown the effectiveness of our method in training large models constrained to our low dimensional manifold. However, as the manifold is modeled as a neural network, the design space of the manifold itself is large. We explore the effect of various design decisions by applying different generator variations to the task of MNIST classification. We intentionally choose a very small dataset so that we can afford running several experiments. Full details of our experimental setup are found in section A.4.

**Effect of choice of activation function:** In section 3.1, we show that for $k = 1$ and $d = 3$, a randomly initialized MLP with sinusoid activations can effectively cover $\mathbb{S}^{d-1}$. For this reason, we selected Sine as our activation function. It's worth asking, however, if this coverage is truly important and whether other activation functions may actually provide better performance. We explore this in Table 5 by comparing against ReLU, Leaky ReLU, ELU, Sigmoid, and no activation function. For the no activation experiment, we no longer train a separate amplitude as the magnitude of the inputs directly control the output magnitudes. Instead, we add this additional parameter as

Table 5: **The impact of MCNC activation function** on MNIST classification.

| Activation Function | Acc. |
|---|---|
| None (linear) | $81.6 \pm 0.5$ |
| ReLU | $76.0 \pm 2.8$ |
| Leaky ReLU | $76.9 \pm 1.6$ |
| ELU | $81.3 \pm 0.2$ |
| Sigmoid | $83.7 \pm 0.7$ |
| Sine | $\mathbf{84.6 \pm 0.7}$ |

an additional input to the generator. From the results, it can be seen that many activation function choices are actively harmful and perform worse than none. Sigmoid and Sine activations are the only ones that outperform none with Sine performing the best. Importantly, when no activation is used, our method recovers a variation of PRANC (Nooralinejad et al., 2023).

**Effect of generator input frequency:** As shown in Section 3.1, the magnitude of the input to the generator, i.e., $L$, controls the number of twists and turns in the manifold. While this twisting is necessary to effectively cover a high dimensional space, it is also likely to increase optimization difficulty. It's worth asking then how different settings of this value affect compression of downstream models. To test this, we multiply the inputs to the generator by a constant associated with the frequency and show results in Table 6. It's immediately apparent that higher frequencies are necessary to achieve high performance as a frequency value of $1.0$ performs similarly to a linear generator. Small increases in frequency greatly increase performance until saturation at $4.0$.

Table 6: **The effect of frequency of first layer of Sine activations** on the accuracy of MNIST.

| Input Frequency | Acc. |
|---|---|
| 1.0 | $81.9 \pm 0.4$ |
| 2.0 | $83.5 \pm 0.2$ |
| 4.0 | $85.1 \pm 0.2$ |
| 8.0 | $84.7 \pm 0.4$ |
| 16.0 | $85.0 \pm 0.4$ |
| 32.0 | $85.5 \pm 0.4$ |

**Varying model size with fixed number of compressed parameters:** A unique aspect of MCNC is that the total number of trainable parameters can be fixed regardless of the number of model parameters by varying its compression rate. This allows us to study how performance scales as model complexity increases while the number of trainable parameters remains the same. As model complexity increases, the number of good solutions should increase. Thus, we anticipate that this over-parameterization will facilitate finding high performing compressed models regardless of the number of trainable parameters. We varied the MLP hidden size while maintaining the number of trainable parameters from the previous experiments and

Table 7: **Impact of increasing model size with fixed number of compressed parameters** on MNIST classification.

| Hidden Size | Acc. |
|---|---|
| 16 | $81.1 \pm 0.3$ |
| 32 | $82.5 \pm 1.1$ |
| 64 | $84.0 \pm 0.7$ |
| 128 | $83.9 \pm 1.3$ |
| 256 | $84.6 \pm 0.4$ |
| 512 | $85.2 \pm 0.2$ |

present the results in Table 7. As expected, we see a consistent increase in accuracy as model size

increases. While smaller architectures can occasionally achieve higher accuracy, the high standard deviations indicate that their optimization often fails to find good solutions.

**Speedup of CPU to GPU transfer:** By decreasing the overall size of the model, MCNC can also be used to reduce the time taken to move large models from CPU to GPU. As long as the generator is loaded into GPU memory, only the much smaller set of $\alpha$s must be transferred from CPU to GPU and then expanded into the full model using the generator. To show this can result in a speedup, we compare the time to load ViT-S from CPU to GPU memory with the time to load and expand the $\alpha$s associated with a ViT-S compressed by 100x. We perform this experiment 100 times using an RTX A6000 and report the average timings in Table 8. These results show that, despite having to complete a forward pass of the generator, we are still able to reduce transfer time by half.

Table 8: **CPU to GPU transfer time of uncompressed/compressed ViT-S model (100x).**

| Uncompressed | Compressed | Speedup |
|---|---|---|
| 35.5 ms | 17.8 ms | 2.0x |

**Effect of training on the generator:** In Section 3.1, we demonstrated that training a generator only slightly improves its coverage of the hypersphere. To further investigate the impact of using trained generators, we applied MCNC with both random and trained generators on CIFAR-10/100. The results presented in Table 9 show consistent but only marginal improvements in accuracy when using trained generators. Given the advantages of random generators, including: 1) easy scalability to different compression rates, 2) storage and communication efficiency (only requiring a random seed), and 3) potential cryptographic benefits, we used random generators for our main experiments.

Table 9: **Impact of using a random vs. trained generator** on CIFAR10/100 classification with different models.

| Arch. | Dataset | Acc. Random | Acc. Trained |
|---|---|---|---|
| R20 | C10 | $82.3 \pm 0.3$ | $82.4 \pm 04$ |
| R56 | C10 | $78.7 \pm 0.2$ | $79.2 \pm 02$ |
| R20 | C100 | $34.7 \pm 0.2$ | $36.1 \pm 0.6$ |
| R56 | C100 | $35.2 \pm 0.1$ | $36.7 \pm 0.5$ |

## 5 CONCLUSION AND LIMITATIONS

Leveraging the prevalence of viable solutions in training overparameterized deep models, we have developed a nonlinear reparameterization method aimed at exploring local minima within a low-dimensional manifold of the parameter space. This approach utilizes a generator model that maps a low-dimensional uniform distribution to a high-dimensional uniform distribution, effectively wrapping the low-dimensional input subspace around the high-dimensional hypersphere. Through extensive experiments in computer vision and NLP tasks, we demonstrate that our method, MCNC, significantly outperforms state-of-the-art baselines in terms of accuracy and/or model reconstruction time. We believe this technique can enhance the storage and deployment of large models, which are integral to the recent advancements in AI, thus facilitating more democratized access to AI tools. However, this may result in negative impacts by putting AI in the hands of many non-sophisticated adversaries.

**Limitations:** We have not fully explored the optimal generator model configuration, including its architecture, loss function, and types of regularizers, and their effects on the downstream optimization. Moreover, due to computational constraints, we have not shown the effect of MCNC in training large language models from scratch, which may reveal the most important impact of MCNC.

ACKNOWLEDGEMENT

This work was partially supported by DARPA under Contract No. HR00112190135 and HR00112290115 and NSF grants 1845216 and 2339898, and NSF CAREER Award 2339898.

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

# A APPENDIX

## A.1 EXAMPLE CODE FOR APPLYING MCNC

Below, we include PyTorch code which shows how to create a linear layer which is reparameterized using MCNC.

```python
import torch
import torch.nn as nn
import torch.nn.functional as F
from math import ceil

class MCNC_Linear(nn.Linear):
    def __init__(
            self,
            generator,
            in_features,
            out_features,
            bias=True):
        """
        Reparameterizes a linear layer with MCNC.

        Args:
            generator (nn.Module): The frozen generator network.
            in_features (int): Number of input features.
            out_features (int): Number of output features.
            bias (bool): If True, includes a bias term.
        """
        super(MCNC_Linear, self).__init__(
            in_features,
            out_features,
            bias)
        self.weight.requires_grad = False
        self.generator = generator

        # Get input and output dimensionality of Generator
        self.k = self.generator.get_input_size()
        self.d = self.generator.get_output_size()

        # Calculate number of Generator inputs to cover the weights
        total_uncompressed_params = torch.numel(self.weight)
        n_gen_inputs = int(ceil(total_uncompressed_params / self.d))

        # Initialize learnable MCNC parameters
        alpha_init = torch.zeros(n_gen_inputs, self.k)
        self.alpha = torch.nn.Parameter(alpha_init, requires_grad=True)

        beta_init = torch.ones(n_alpha, 1)
        self.beta = torch.nn.Parameter(beta_init, requires_grad=True)

    def forward(self, x):
        """
        Forward pass reparameterizing weights with MCNC.

```

```
48          Args:
49              x (torch.Tensor): Input data.
50
51          Returns:
52              torch.Tensor: Output of the linear layer.
53          """
54          # Reparameterize weight with MCNC
55          gen_out = self.generator(self.alpha) * self.beta
56          w = gen_out.view(self.weight.shape) + self.weight
57          return F.linear(x, w, self.bias)
```

## A.2   RECOMMENDED SETTINGS FOR MCNC

Table 10: **Recommended hyperparameter defaults** for MCNC.

| Setting | Recommendation |
|---|---|
| Input Dimension | 9 |
| # of layers | 3 |
| Width | 1000 |
| Input Frequency | 4.5 |
| Weight Initialization | $U([-\frac{1}{n}, \frac{1}{n}])$ |
| Optimizer | Adam |
| Learning Rate | 5-10x larger than uncompressed model |

## A.3   DETAILS ON COMPRESSING IMAGE CLASSIFIERS

**Generator architecture** When training from scratch with MCNC, we use the same Generator configuration as listed in Table 10 and set $d$ such that we achieve the desired compression. All linear layers in our generator do not include a bias so that we guarantee a zero initialization by setting our inputs to zero.

**ViT hyperparameters** The baseline models used for pruning are trained for 800 epochs using the AdamW (Loshchilov & Hutter, 2019) optimizer. We set batch size to 1024 with initial learning rate of 0.001 and use a cosine learning rate scheduler. We use the same augmentations oulined in (Touvron et al., 2021) for ImageNet. For all pruning methods, we search for the best learning rate in $\{5e-4, 1e-4, 8e-5\}$ per compression rate. We train for 20 epochs and set the parameters of the cubic sparsity scheduler to $t_i = 1000, t_f = 12000$. For PLATON, we set $\beta_1 = 0.85$ and $\beta_2 = 0.95$. In addition, for each compression rate we run both with and without MixUp (Zhang et al., 2018) and CutMix (Yun et al., 2019) augmentations and take the best results. We use the AdamW (Loshchilov & Hutter, 2019) optimizer and a cosine learning rate scheduler. For MCNC, we train for 800 epochs with a cosine learning rate scheduler using the Adam (Kingma & Ba, 2015) optimizer. Learning rates and whether Mixup/Cutmix are used for each method are shown in Tables 11 and 12. For pruning methods, we set batch size to 128 and use 256 for MCNC on ViT-Ti and 512 for MCNC on ViT-S.

**ResNet hyperparameters** For all experiments, we use the same set of data augmentations as in PRANC (Nooralinejad et al., 2023). Likewise, we exclude BatchNorm (Ioffe & Szegedy, 2015) parameters from our compression and do not consider them when computing the compression rate. When applying MCNC to ResNet architectures, we train for 400 epochs with a learning rate of 0.01 using the Adam (Kingma & Ba, 2015) optimizer. We decay the learning rate by 0.5 if the loss has not improved in 4 epochs. When applying LoRA to a convolutional layer with kernel size $k$ and input and output channels $c_1$ and $c_2$, we reshape it be of size $kc_1 \times kc_2$ and create low rank matrices $A \in \mathbb{R}^{kc_1 \times r}, B \in \mathbb{R}^{r \times kc_2}$. When applying LoRA, we use rank 64 as in (Koohpayegani et al., 2024) and do not attempt to find the optimal rank.

## A.4   ABLATION EXPERIMENTAL SETUP

Unless stated otherwise, all ablation experiments follow the same experimental setup. In general, we compress an MLP with two hidden layers and hidden size 256 compressed to only $0.2\%$ of the original parameters. We choose such an extreme compression so that the problem is difficult enough

Table 11: **Hyperparameters for compressing ViT-Ti**

| Method | Percentage of Model Size | Learning Rate | Mixup/Cutmix Augmentations |
|---|---|---|---|
| Magnitude | 50% | 0.0005 | ✓ |
| PLATON | 50% | 0.0005 | ✓ |
| MCNC | 50% | 0.001 | ✓ |
| Magnitude | 20% | 0.0005 | × |
| PLATON | 20% | 0.0005 | × |
| MCNC | 20% | 0.001 | ✓ |
| Magnitude | 10% | 0.0005 | × |
| PLATON | 10% | 0.0005 | × |
| MCNC | 10% | 0.01 | × |
| Magnitude | 5% | 0.0005 | × |
| PLATON | 5% | 0.0005 | × |
| MCNC | 5% | 0.01 | × |
| Magnitude | 2% | 0.0005 | × |
| PLATON | 2% | 0.0005 | × |
| MCNC | 2% | 0.01 | × |
| Magnitude | 1% | 0.0005 | × |
| PLATON | 1% | 0.0005 | × |
| MCNC | 1% | 0.01 | × |

Table 12: **Hyperparameters for compressing ViT-S**

| Method | Percentage of Model Size | Learning Rate | Mixup/Cutmix Augmentations |
|---|---|---|---|
| Magnitude | 25% | 0.0005 | ✓ |
| PLATON | 25% | 0.0005 | ✓ |
| MCNC | 25% | 0.002 | ✓ |
| Magnitude | 10% | 0.0005 | ✓ |
| PLATON | 10% | 0.0005 | × |
| MCNC | 10% | 0.002 | ✓ |
| Magnitude | 5% | 0.0005 | × |
| PLATON | 5% | 0.0005 | × |
| MCNC | 5% | 0.005 | ✓ |
| Magnitude | 2% | 0.0005 | × |
| PLATON | 2% | 0.0005 | × |
| MCNC | 2% | 0.005 | × |

to see variations in design decisions. Note that while the reported numbers of $\sim 85\%$ accuracy are low for MNIST classification, training a linear classifier with $\ell_1$ regularization and sparsifying to a similar number of parameters results in $\sim 79\%$ accuracy. The default architecture of our generator has an input size of 9, two hidden layers of size $1,000$, and an output size of $5,000$. Note that $\frac{9+1}{5,000} = 0.002$. For each experiment, we search for the best learning rate in $\{0.1, 0.01, 0.001\}$ and report results averaged across three trials.

Table 15: **The effect of generator hidden size** on the accuracy of MNIST.

| Generator Width | Acc. |
|---|---|
| 64 | $83.5 \pm 0.8$ |
| 128 | $84.3 \pm 0.4$ |
| 256 | $84.3 \pm 0.2$ |
| 512 | $84.7 \pm 0.8$ |
| 1024 | $84.5 \pm 0.5$ |
| 2048 | $85.0 \pm 0.7$ |

Table 16: **The effect of generator depth** on the accuracy of MNIST.

| # of Layers | Accuracy w/o Residual Conn. | Accuracy w/ Residual Conn. |
|---|---|---|
| 2 | $83.7 \pm 0.9$ | N/A |
| 3 | $84.9 \pm 0.8$ | $83.9 \pm 0.5$ |
| 4 | $85.2 \pm 0.1$ | $84.0 \pm 0.3$ |
| 5 | $85.2 \pm 0.8$ | $84.0 \pm 0.5$ |

### A.5 ADDITIONAL ABLATION EXPERIMENTS

**Impact of varying $k$ and $d$ with fixed compression rate:** The ratio of $k$ to $d$ determines its compression rate. Here, we multiply the input and output size of the random generator by an increasing constant factor keeping the compression rate fixed. The results are presented in Table 13. It is evident that a $k$ value close to 1 leads to poor performance. As each generator input has an associated amplitude, the amplitude uses a larger percentage of the learnable parameters for smaller values of $k$. As $k$ increases, the amplitude uses a much smaller percentage of the parameter budget, increasing the generator's ability to model complex functions.

Table 13: **Impact of varying generator input and output size with a fixed compression rate** on MNIST classification.

| $k$ | $d$ | Acc. |
|---|---|---|
| 1 | 1000 | $69.3 \pm 1.8$ |
| 3 | 2000 | $78.4 \pm 0.2$ |
| 7 | 4000 | $83.2 \pm 0.7$ |
| 15 | 8000 | $84.9 \pm 0.2$ |
| 31 | 16000 | $\mathbf{85.8 \pm 0.2}$ |

**Effect of Generator weight initialization** By representing our manifold as a randomly initialized neural network, it is able to easily be communicated using only a random seed. However, the space of possible random initialization schemes is large, and it is not clear what method of initialization would be most effective. To study this, we initialize generators where the weights are generated from both normal and uniform distributions with multiplying the variance of the distribution by a factor of $c$. As this multiplication also implicitly controls the input frequency, we always let $c = 1$ for the first layer. We present results of this experiment in Table 14. It's clear from these results that drawing weights from a uniform distribution provides better performance than a normal distribution. In addition, results are higher when the variance is smaller. We note, however, that these results are likely intertwined with the selection of frequency in the first layer.

Table 14: **The effect of generator weight initialization** on the accuracy of MNIST. We vary the scale of each distribution by multiplying the variance by $c$.

| Initialization | $c$ | Acc. |
|---|---|---|
| Uniform | 0.5 | $\mathbf{85.1 \pm 0.2}$ |
| Uniform | 1.0 | $84.6 \pm 0.0$ |
| Uniform | 2.0 | $83.2 \pm 0.9$ |
| Uniform | 4.0 | $80.6 \pm 1.6$ |
| Uniform | 8.0 | $80.1 \pm 0.8$ |
| Normal | 0.5 | $81.8 \pm 0.5$ |
| Normal | 1.0 | $81.6 \pm 0.2$ |
| Normal | 2.0 | $82.1 \pm 0.9$ |
| Normal | 4.0 | $81.9 \pm 0.2$ |
| Normal | 8.0 | $82.0 \pm 1.0$ |

**Effect of generator width and depth** It's unclear how architecture variations in the generator of MCNC affect compression performance. To see the effect of generator hidden size, we construct generators with three layers and vary the hidden dimension and show results in Table 15. While accuracy initially improves, it quickly saturates suggesting there is not a benefit to infinitely increasing width. Likewise, we study the effect of depth by constructing generators with a hidden size of 1000 and varying the number of layers. In addition, we experiment with the effect of residual connections as it's possible that increased depth causes our optimization to experience vanishing or exploding gradients. We present the results in Table 16. Similarly, we see an improvement when extending beyond a single hidden layer.

### A.6 COMPUTATION OF RECONSTRUCTION FLOPS FOR LLAMA-2 7B AND 13B

For Llama-2-7b, note that the model is composed of 32 layers. Each transformer layer can then be decomposed into 7 linear layers which we apply our method to:

1. 4 linear layers in the self-attention block each of size $4096 \times 4096$
2. 3 linear layers in the MLP block. Each has size $4096 \times 11008$. Note that the third layer comes from the gate of the SwiGLU activation function (Shazeer, 2020)

We use rank 8 for both NOLA and our method which gives 11 matrices of size $4096 \times 8$ and 3 matrices of size $11008 \times 8$. Assuming we can compute the flops to generate each of these matrices then the total flops would be:

$$32 * (11 * \text{FLOPS}(4096 \times 8) + 3 * \text{FLOPS}(11008 \times 8))$$

For NOLA, we use 64 basis so:

$$\text{FLOPS}(4096 \times 8) = 2 * 64 * 4096 * 8 = 4.19\text{MFLOPS}$$
$$\text{FLOPS}(11008 \times 8) = 2 * 64 * 11008 * 8 = 11.27\text{MFLOPS}$$

Giving us a total of: $32 * (11 * 4.19 + 3 * 11.27) = 2.56\text{GFLOPS}$

For MCNC, each run of the generator requires $2 * (5 * 32 + 32 * 32 + 32 * 5000)$ operations. The number of forward passes necessary for each matrix is:

$$ceil(\frac{4096 \times 8}{5000}) = 7$$
$$ceil(\frac{11008 \times 8}{5000}) = 18$$

We also need to multiply each 5000 dimensional output by a single scalar, so the total number of operations is:

$$\text{FLOPS}(4096 \times 8) = 7 * 2 * (5 * 32 + 32 * 32 + 32 * 5000) + 7 * 5000 = 2.29\text{MFLOPS}$$
$$\text{FLOPS}(11008 \times 8) = 18 * 2 * (5 * 32 + 32 * 32 + 32 * 5000) + 18 * 5000 = 5.89\text{MFLOPS}$$

Giving us a total of: $32 * (11 * 2.29 + 3 * 5.89) = 1.37\text{GFLOPS}$

For Llama-2-13b, the computation is the same except:

1. there are 40 layers
2. hidden dimension is 5120
3. intermediate size is 13824
4. LoRA rank is 16

