# OpenReview forum: "MCNC: Manifold-Constrained Reparameterization for Neural Compression"
_ICLR.cc/2025/Conference — ICLR 2025 Poster_

### Official Review · Reviewer_dj4q · 2024-11-03

**Soundness:** 3
**Presentation:** 2
**Contribution:** 3
**Rating:** 6
**Confidence:** 4

**Summary:**

The paper proposes a reparameterization to train neural networks which leads to a lower memory footprint for transferring and storing the parameters. The paper achieves this by re-parameterizing the network parameters to lie on a lower dimensional spherical manifold. The method uses a “generator” network for this, mapping k-dimensional vectors onto a d-dimensional sphere. The k-dimensional vectors are then trained. The paper compares this method against multiple model compression and efficient training methods, demonstrating performance gains.

**Strengths:**

1. Strong empirical results, with comparisons against other compression methods on various modalities and architectures demonstrating the proposed method’s effectiveness.
2. Extensive ablation studies for each component of the method, as well as for choices of hyper-parameters, giving some intuition on how
3. Demonstration of real world speedups

**Weaknesses:**

1. My main gripe with the paper is that the method is not very well motivated. For example, the paper claims that a randomly initialized generator is enough to span the entire space of weights in a higher dimension, but never shows much empirical evidence beyond an experiment in low dimensions. It is not clear if this result also holds in higher dimensions. I would have ideally liked to see a theoretical result around the expressivity of the manifold constrained reparametrization, but even an empirical investigation showing that one does not lose out on expressivity would be great.
2. Continuing on my previous point, from table 10, it appears that a trained generator is better for larger models. However, this result is down-played in the text description, with the narrative claiming that the improvement is "marginal".

**Questions:**

1. How was the wasserstein distance computed for Fig 2? If I understand correctly, there is no closed form expression to compute this metric in higher dimensions.
2. Why does ViT-S have better accuracy for smaller models in Tab 1? Is it an artifact of high variance in the results?
3. How were reconstruction GFLOPs computed for Table 4?
4. How much slower is training MCNC as compared to baseline LoRA?

---

> ### Author Response · Authors · 2024-11-22
>
> We thank the reviewer for their time and consideration. Below please see our answers to the raised questions.
>
> > the paper claims that a randomly initialized generator is enough to span the entire space of weights in a higher dimension, but never shows much empirical evidence beyond an experiment in low dimensions. It is not clear if this result also holds in higher dimensions. I would have ideally liked to see a theoretical result around the expressivity of the manifold constrained reparametrization, but even an empirical investigation showing that one does not lose out on expressivity would be great.
>
> We first wish to clarify that we do not expect a randomly initialized generator to span the entire space of weights in a higher dimension. This is infeasible without assuming access to infinite precision values. Otherwise, we are inherently limited by the quantization of the input space. Instead, we claim that a randomly initialized generator is not significantly worse than a trained generator. We do agree, however, that we were not thorough in our evaluation of this claim. To further evaluate our claim, we provide additional comparisons between random initialization and trained generators for higher dimensions for $k=2$ below:
>
> | k| d | W2 Random | W2 Trained | W2 Dirac | W2 $U(S^{d-1})$ |
> | -------- | -------- | -------- | -------- | -------- | -------- |
> | 2 | 4 | 0.0703 (90.6%) | 0.0066 (99.9%) | 0.7039 (0.0%) | 0.0034 (100.0%)  |
> | 2 | 8 | 0.0305 (96.3%) | 0.0035 (99.9%) | 0.4971 (0.0%) | 0.0027 (100.0%)  |
> | 2 | 16 | 0.0171 (97.8%) | 0.0033 (99.9%) | 0.3521 (0.0%) | 0.0022 (100.0%)  |
> | 2 | 32 | 0.0173 (95.6%) | 0.0024 (99.9%) | 0.2488 (0.0%) | 0.0012 (100.0%)  |
>
> We compute the above using the sliced wasserstein distance with 100,000 samples and 10,000 slices. We use the sliced-Wasserstein distance instead of Wasserstein here due to its: 1) superior computational speed, and 2) better sample complexity. We use the same set of target samples and slices for each to ensure a fair comparison. We also include the distance between a random dirac distribution on the sphere as well as a separate set of 100,000 points sampled from the ground truth distribution to further contextualize the reported distances. Note that the decrease in distance as dimensionality increases is due to the likelihood of sampling an informative slice decreasing exponentially with respect to dimensionality. In addition to the sliced-wasserstein distance, we also report $100*\exp(-\tau SW_2^2(\hat{\mu}, \nu))$ where $\tau$ is set such that the value of the dirac distribution is less than 1e-2. Lastly, we emphasize that, due to the over-parameterization of neural networks, full coverage of high dimensional space might not be necessarily needed.
>
> > Continuing on my previous point, from table 10, it appears that a trained generator is better for larger models. However, this result is down-played in the text description, with the narrative claiming that the improvement is "marginal"
>
> While a trained generator does show an improvement over our randomly initialized generator, it is our opinion that the degradation is not too severe given the additional benefits of using a random initialization. By using a random initialization, we are able to communicate/store the generator using only a random seed. If we were to use a trained generator instead, it would be necessary to either include the entire generator in the compression budget or assume the generator has been pre-installed in the downstream device. Additionally, a randomly initialized generator allows greater flexibility as it allows generators of varying compression rates to be easily generated on demand. However, we do agree that if an application can utilize a trained generator, it is preferable to do so.
>
> > How was the wasserstein distance computed for Fig 2? If I understand correctly, there is no closed form expression to compute this metric in higher dimensions.
>
> You are correct that there is no closed-form expression for computing the Wasserstein distance between arbitrary distributions. In our approach, we randomly sample $N$ points from a uniform distribution over $[-L, L]$, pass them through the generator to produce samples from its output distribution, and compare these to $N$ points sampled uniformly from the sphere (obtained by sampling from a $d$-dimensional normal distribution and normalizing the samples). The Wasserstein distance between these two empirical distributions is then computed using linear programming. However, the computational complexity of this approach becomes prohibitive for large $N$ due to the cubic complexity of linear programming. Alternatively, given the improved sample efficiency of the sliced Wasserstein distance, we opt to report results using this metric, following the procedure outlined above.

---

> > ### Author Response · Authors · 2024-11-22
> >
> > > Why does ViT-S have better accuracy for smaller models in Tab 1? Is it an artifact of high variance in the results?
> >
> > Thanks to the reviewer's feedback, we refined the codebase for this experiment, reran the experiments for ViT-S, and have updated the numbers in the submission. The new results show that we can recover baseline model performance while consistently out-performing the compared pruning methods. The paper and the table are revised accordingly.
> >
> > > How were reconstruction GFLOPs computed for Table 4?
> >
> > For Llama-2-7b, note that the model is composed of 32 layers. Each transformer layer can then be decomposed into 7 linear layers which we apply our method to:
> > - 4 linear layers in the self-attention block each of size $4096\times4096$
> > - 3 linear layers in the MLP block. Each has size $4096\times11008$
> > -- The third linear layer comes from the gate of the SwiGLU activation function[1].
> >
> > We use rank 8 for both NOLA and our method which gives:
> > - 11 matrices of size $4096\times8$
> > - 3 matrices of size $11008\times8$
> >
> > Assuming we can compute the flops to generate each of these matrices then the total flops would be:
> > $$32*(11 * FLOPS(4096\times8) + 3 * FLOPS(11008\times8))$$
> >
> > For NOLA, we use 64 basis so:
> > $$FLOPS(4096 \times 8) = 2 * 64 * 4096 * 8 = 4.19\text{MFLOPS}$$
> >
> > $$FLOPS(11008\times8) = 2 * 64 * 11008 * 8 = 11.27\text{MFLOPS}$$
> >
> > Giving us a total of: $32 * (11 * 4.19 + 3 * 11.27) = 2.56\text{GFLOPS}$
> >
> > For MCNC, each run of the generator requires $2 * (5 * 32 + 32 * 32 + 32 * 5000)$ operations. The number of forward passes necessary for each matrix is:
> >
> > $$ceil(\frac{4096 \times 8}{5000})=7$$ $$ceil(\frac{11008 \times 8}{5000})=18$$
> >
> > We also need to multiply each 5000 dimensional output by a single scalar, so the total number of operations is:
> > $$FLOPS(4096\times8) = 7 * 2 * (5 * 32 + 32 * 32 + 32 * 5000) + 7 * 5000 = 2.29\text{MFLOPS}$$ $$FLOPS(11008\times8) = 18 * 2 * (5 * 32+32 * 32+32 * 5000) + 18 * 5000 = 5.89\text{MFLOPS}$$
> >
> > Giving us a total of: $32 * (11 * 2.29 + 3 * 5.89) = 1.37\text{GFLOPS}$
> >
> > For Llama-2-13b, the computation is the same except:
> > - there are 40 layers
> > - hidden dimension is 5120
> > - intermediate size is 13824
> >
> > Finally, we realized while recreating the numbers that the FLOPS for NOLA on LLaMA-2 13B were computed using a rank of 8 instead of 16. We have corrected this in the submission.
> >
> > > How much slower is training MCNC as compared to baseline LoRA?
> >
> > We provide the total training time for LoRA, NOLA, and MCNC for LLaMA-2 7B and 13B below:
> >
> > 7B:
> > - LoRA: 121 minutes
> > - NOLA: 176 minutes (45% increase)
> > - MCNC: 132 minutes (9% increase)
> >
> > 13B:
> > - LoRA: 134 minutes
> > - NOLA: 229 minutes (71% increase)
> > - MCNC: 148 minutes (10%)
> >
> >
> > [1]: Noam Shazeer, "GLU Variants Improve Transformer," arXiv, 2020. [arXiv:2002.05202](https://arxiv.org/abs/2002.05202).

---

> ### Comment · Reviewer_dj4q · 2024-11-26
> **Official Comment (1/2)**
>
> I thank the authors for their response.
> > FLOP computations
>
> The computation of FLOPs makes it clear as to exactly where the savings come for MCNC on top of NOLA. I would encourage the authors to add this to the appendix for better exposition.
>
> > Accuracy on ViT-S
>
> I am curious as to what changed in the experiments to produce this version of the table. Here, it appears that MCNC is much better than all the baselines, and more interestingly, the 25% sized model produced by MCNC is also better than the 100% trained base model! I would appreciate any insights of the authors here.
>
> > Training Time
>
> I appreciate that the method has only small overheads above LoRA despite strong performance and much smaller memory overhead.

---

> > ### Author Response · Authors · 2024-11-26
> >
> > > The computation of FLOPs makes it clear as to exactly where the savings come for MCNC on top of NOLA. I would encourage the authors to add this to the appendix for better exposition.
> >
> > We thank the reviewer for this suggestion which improves the clarity of our submission. We have added an additional section to the appendix with the math from our previous response.
> >
> > > I am curious as to what changed in the experiments to produce this version of the table. Here, it appears that MCNC is much better than all the baselines, and more interestingly, the 25% sized model produced by MCNC is also better than the 100% trained base model! I would appreciate any insights of the authors here.
> >
> > Thank you for your insightful observation regarding the improved performance of MCNC in the updated table. In the earlier version of the table, a version control issue resulted in a discrepancy in the data augmentation process when applying MCNC to ViT-S. Specifically, only random horizontal flip combined with MixUp/CutMix augmentations was used. The updated table now applies the same comprehensive set of augmentations used for training ViT-Ti, including random erasing, color jittering, and Rand-Augment. Importantly, this adjustment affected only the MCNC entries for ViT-S, as its training code operates independently from other methods. The revised results more accurately represent MCNC's performance, particularly for the larger ViT-S model, where proper augmentations have a substantial impact on training the model.
> >
> > The improved accuracy of the 25% compressed model compared to the baseline 100% model can be attributed to the implicit regularization introduced by our proposed reparameterization. This regularization helps mitigate overfitting, which is particularly beneficial given the relatively small size of the ImageNet-100 dataset. We emphasize that all our experiments follow standard practices to mitigate overfitting during model training, including the application of weight decay as a regularization technique. The table below illustrates both the training and testing accuracy of the baseline and MCNC for 25% compression:
> >
> > | Method | train acc. | test acc.|
> > | -------- | -------- | -------- |
> > | Baseline (0% Compression) | 82.5% | 78.8% |
> > | MCNC (25% Compression) | 78.8% | 79.2% |
> >
> > As shown, MCNC achieves slightly lower training accuracy at 25% compression but outperforms in test accuracy, indicating a reduced generalization gap. This outcome underscores the potential of MCNC's regularization mechanism when learning from smaller datasets like ImageNet-100.

---

> > > ### Comment · Reviewer_dj4q · 2024-11-26
> > > **Official Response**
> > >
> > > I thank the authors for their response.
> > >
> > > This addresses concerns about the empirical performance of the method and I am raising my rating based on this.
> > >
> > > There is still some doubt in my mind around the earlier discussion around the Wasserstein distance and expressivity of the method that I will write a more detailed comment about soon.

---

### Official Review · Reviewer_5SSn · 2024-11-03

**Soundness:** 4
**Presentation:** 2
**Contribution:** 3
**Rating:** 6
**Confidence:** 4

**Summary:**

The paper presents a new approach to parametrize the network parameters: via another generator model. Specifically, weights are split into blocks(vectors) of size $d$ (i.e., $\mathbb{R}^d$) and parametrized to be coming from a $k$-dimensional space ($\mathbb{R}^k$) via mapping $\phi: \mathbb{R}^k \to \mathbb{R}^d $. The mapping $\phi$ is defined as a compact feed-forward neural network with sine activation with number of layers and hidden dimensions being hyper-parameters of choice; though for most experiments there are not more 3 layers with hidden dimensions of not more 1024. The mapping $\phi$ is randomly initialized, and fixed. The only trainable parameters are the actual inputs to the $\phi$, i.e., the inverted image of parameters in this lower-dimensional space. The authors demonstrate that such a simple setup allows to train from scratch and finetune various models in parameter efficient way.

**Strengths:**

The paper presents very simple yet powerful idea of essentially hashing high-dimensional weights into lower-dimensional manifold. Instead of hashing directly from higher to lower dimension, the proposed approach fixes the inverts of the hashing function (i.e., generator), and learns the "hashed" outputs via sgd. This idea has deep roots in random projections/random hashing literature, and as such I am very glad that similar approach can be exploited for the parameter efficient training of the neural networks.

**Weaknesses:**

Overall, I think the paper does a good job in validating the soundness and effectiveness of the proposed approach, however, I think it can be significantly strengthened:
1. In the exposition of the idea, the actual architecture of the generator network comes very late; with actual numbers/settings coming in appendix! I think, the simplicity of the approach must be showcased from the beginning, I highly suggest to add simple pytorch code showing how easy is to plug and play with the method
2. Similar to the previous weakness, I think authors should consider recommending some bulletproof settings that would work for most of the cases (like number of layer in $\phi$, number of dimensions, some rules of thumb, how long to train, optimizer settings) to easy any potential adoption: at the end of the day, the ideas have very limited impact if not used in practice.
3. Theoretical guarantees of any sorts would be welcome as well: how universal is such a generator network? can it be used to train/finetune any neural network or it has limitations?
4. While the papers literature review is extensive, I think the connection to hashing must be explored in depth and more papers included into review (some suggestions: Structured Multi-Hashing for Model Compression,  Lossy weight encoding for deep neural network compression)

**Questions:**

Please address weaknesses above.

---

> ### Author Response · Authors · 2024-11-22
>
> We thank the reviewer for their time and comments. Below see our response to the raised questions and concerns.
>
> > In the exposition of the idea, the actual architecture of the generator network comes very late; with actual numbers/settings coming in appendix! I think, the simplicity of the approach must be showcased from the beginning, I highly suggest to add simple pytorch code showing how easy is to plug and play with the method
>
> We thank the reviewer for this suggestion. We have added a short snippet of PyTorch code showing how to perform the reparameterization for a PyTorch Linear layer. Unfortunately, page constraints make it difficult to include this in the main paper so this has been added to the top of the appendix.
>
> > Similar to the previous weakness, I think authors should consider recommending some bulletproof settings that would work for most of the cases (like number of layer in, number of dimensions, some rules of thumb, how long to train, optimizer settings) to easy any potential adoption: at the end of the day, the ideas have very limited impact if not used in practice.
>
> This is a good point. As we use the same generator configuration for all experiments when training from scratch, we have migrated these settings into a table so that readers can quickly find the recommended configuration. In this table, we also include recommendations for optimizer and learning rate.
>
> > Theoretical guarantees of any sorts would be welcome as well: how universal is such a generator network? can it be used to train/finetune any neural network or it has limitations?
>
> This is an area of great interest to us as well. While we do not yet have concrete theoretical results, we believe there may be connections between our work and recent theoretical advancements on gradient descent with large learning rates[1]. By reparameterizing a network as the output of a sinusoid-activated generator, we constrain the weight space to a highly dynamic manifold. This allows small changes in the generator's inputs to induce substantial changes in the model's weights. This behavior suggests potential links between our approach and the theoretical insights into optimization dynamics with large learning rates. These connections and a more theoretical exploration of the MCNC framework will be addressed in future work.
>
> > While the papers literature review is extensive, I think the connection to hashing must be explored in depth and more papers included into review (some suggestions: Structured Multi-Hashing for Model Compression, Lossy weight encoding for deep neural network compression)
>
> We thank the reviewer for drawing our attention to these works. We have added them to our related works section. As hashing methods are heavily related to weight sharing, we have added them to this section.
>
> [1]: Cai et al., "Large Stepsize Gradient Descent for Non-Homogeneous Two-Layer Networks: Margin Improvement and Fast Optimization," arXiv, 2024. [arXiv:2306.07629](https://arxiv.org/abs/2306.07629).

---

### Official Review · Reviewer_97Hi · 2024-11-04

**Soundness:** 4
**Presentation:** 4
**Contribution:** 4
**Rating:** 6
**Confidence:** 4

**Summary:**

The paper proposed a new approach to compress deep neural networks. The basic idea is to enforce the DNN parameters to lie on a k-dimension manifold for every chunk of d parameters where k << d. The proposed approach uses a generator model that maps a low-dimensional uniform distribution (k-dimension) to a high-dimensional uniform distribution (d-dimension). Evaluations on computer vision and NLP tasks demonstrate that the proposed approaches can achieve extremely high compression rates.

**Strengths:**

+ The proposed algorithm outperforms baselines in the high compression rate regions
+ The idea is interesting and shows that there exists a set of parameters in a DNN that lies in the lower-dimension manifold.
+ Evaluations are thorough

**Weaknesses:**

The paper presents an interesting idea of compressing neural network parameters into a low dimension manifold. Instead of training a generator to produce the neural network parameters, the proposed approach can use a randomly-initialized generator to project a trainable k-dimension vector into a d-dimension vector as parameters. I find the idea very interesting. The empirical results also demonstrate that the approach can achieve better accuracy under extreme compression rates.

With that said, here are a few concerns on the work
- The benefits of achieving extreme compression rates under the high accuracy loss are not clear to me. Using table 1's result as an example, with 1% percentage of model size, the proposed approach can achieve an accuracy of 49% compared to the baselines which are 33%-39% on ViT-Ti model. However, compared to the baseline model accuracy 76.2%, it is a huge accuracy drop (more than 20%). Under this context, one might consider using a different model variant that is smaller but achieves a higher accuracy than compressing the ViT-Ti model.  Although the proposed method can achieve better accuracy compared to baselines under high compression rates, I am concerned about how likely such advantages would be useful in practice due to the significant accuracy drop compared to the non-compressed or lightly-compressed model.

- Table 4 highlights the same concern. The trainable parameters from the LoRA model are already minimal compared to the number of parameters in the LLaMA-2 7B and 13B. With the quantization from NOLA and MCNC, both of them slow down the training throughput of the model. I agree with the advantages of MCNC compared to NOLA, but neither approaches are appealing compared to running the original LoRA method.

**Questions:**

Can the author clarify how does the benefit of the proposed approach over baselines translate to practical computational performance gains?

---

> ### Author Response · Authors · 2024-11-22
>
> We thank the reviewer for their time and consideration. Please find questions to your questions and comments below.
>
> > The benefits of achieving extreme compression rates under the high accuracy loss are not clear to me. Using table 1's result as an example, with 1% percentage of model size, the proposed approach can achieve an accuracy of 49% compared to the baselines which are 33%-39% on ViT-Ti model. However, compared to the baseline model accuracy 76.2%, it is a huge accuracy drop (more than 20%). Under this context, one might consider using a different model variant that is smaller but achieves a higher accuracy than compressing the ViT-Ti model. Although the proposed method can achieve better accuracy compared to baselines under high compression rates, I am concerned about how likely such advantages would be useful in practice due to the significant accuracy drop compared to the non-compressed or lightly-compressed model.
>
> The accuracy loss of extreme compression rates are an understandable concern. We first highlight our results at 10% compression rate of ViT-Ti. Here, we show that we are able to outperform pruning methods with performance only dropping ~6% accuracy. While there is significant degradation at the more extreme compression rates, we show that we outperform pruning in this setting. Although training a smaller architecture appears to be a plausible alternative, prior work PRANC (ECCV 2023) showed that their method significantly outperforms a knowledge distilled LeNet (60k parameters) with a compressed ResNet-56 (5k parameters) on CIFAR-10 classification. In our experiments, we show that we out-perform PRANC in a similar setting.
>
> > Table 4 highlights the same concern. The trainable parameters from the LoRA model are already minimal compared to the number of parameters in the LLaMA-2 7B and 13B. With the quantization from NOLA and MCNC, both of them slow down the training throughput of the model. I agree with the advantages of MCNC compared to NOLA, but neither approaches are appealing compared to running the original LoRA method.
>
> When compared to the number of parameters in LLaMA-2 7B and 13B, the number of LoRA parameters is indeed small. However, despite being relatively smaller, the parameter count for LoRA remains significant. For example, implementing LoRA at rank=1 for LLaMA-2 7B requires 2.5M parameters, and for LLaMA-2 13B, it requires 3.9M parameters. Typically, higher ranks are needed for effective PEFT with LoRA, which can pose challenges depending on the application. For instance, consider a scenario where an LLM must be personalized for each user of a system. To prevent private data leakage, a reasonable solution would involve fine-tuning the LLM independently for each user. Supporting 1,000 users on LLaMA-2 13B under this approach would demand an additional 3.9GB of storage (assuming rank=1). In contrast, MCNC would require only 77MB for the same use case.
>
> > Can the author clarify how does the benefit of the proposed approach over baselines translate to practical computational performance gains?
>
> As MCNC directly generates model parameters, it does not provide performance improvements in terms of FLOPS. However, MCNC can significantly decrease a model's memory footprint. This reduction is particularly impactful in scenarios where a model is too large to fit within GPU memory, as the time required to transfer model parameters from CPU to GPU can dominate inference latency. This issue is already a significant bottleneck for LLMs. For instance, the recently proposed quantization method, SqueezeLLM[1], specifically targets memory optimization to improve inference latency rather than computation. Beyond LLMs, memory-constrained devices can benefit from running compressed versions of larger models. Additionally, applications that require hosting multiple models on a single GPU can leverage MCNC's improved CPU-to-GPU transfer times, as demonstrated in Table 9.
>
> [1]: Kim et al., "SqueezeLLM: Dense-and-Sparse Quantization," arXiv, 2023. [arXiv:2306.07629](https://arxiv.org/abs/2306.07629).

---

> ### Comment · Reviewer_97Hi · 2024-11-27
>
> I appreciate the authors' comments and responses to my questions. However, my question regarding the extent of the benefits provided by the proposed approach still stands. The citation to SqueezeLLM is not particularly relevant in this context. Note that while SqueezeLLM focuses on compressing the LLM model itself, this work compresses the LoRA parameters within an LLM. The memory pressure caused by an LLM with 13 billion parameters is far more apparent than one caused by just a few million LoRA parameters, making the latter scenario less compelling by comparison. With the additional computation overheads from generating the parameters, it is not obvious that the approach will bring inference speedups, even if memory is a bottleneck.
>
> With that said, I do see a compelling case when the approach is applied to the entire model with a high compression rate (e.g., ResNets, ViT). The compressed models can fit into resource-constrained devices with limited memory.

---

> > ### Author Response · Authors · 2024-11-29
> >
> > > I appreciate the authors' comments and responses to my questions. However, my question regarding the extent of the benefits provided by the proposed approach still stands. The citation to SqueezeLLM is not particularly relevant in this context. Note that while SqueezeLLM focuses on compressing the LLM model itself, this work compresses the LoRA parameters within an LLM. The memory pressure caused by an LLM with 13 billion parameters is far more apparent than one caused by just a few million LoRA parameters, making the latter scenario less compelling by comparison. With the additional computation overheads from generating the parameters, it is not obvious that the approach will bring inference speedups, even if memory is a bottleneck.
> >
> > We thank the reviewer for their insightful comment and appreciate the opportunity to clarify. We agree with the reviewer that the *SqueezeLLM* paper does not focus on PEFT but compresses the entire model. The central takeaway of that paper, as noted, is that "the main bottleneck for generative inference with LLMs is memory bandwidth, rather than compute." Below we show that MCNC helps with this.
> >
> > We emphasize that our proposed framework, MCNC, supports both full network compression, as demonstrated for ViT models, and the compression of fine-tuning parameters. However, due to the training requirements of MCNC, which involves manifold-constrained optimization, we currently lack the computational resources to train an LLM from scratch using this method. Therefore, we limited our experiments to the PEFT setting, which is less computationally demanding.
> >
> > The reviewer raises an excellent point regarding the potential inference overhead introduced by generating parameters: "With the additional computation overheads from generating the parameters, it is not obvious that the approach will bring inference speedups, even if memory is a bottleneck." To address this, we conducted an experiment comparing the inference time of a LLaMA-65B model with and without MCNC. We use a single sample of sequence length 256 for this experiment. As loading the entire LLaMA-65B model, even on the CPU, is taxing, we simulate the model by constructing enough LLaMA-65B transformer layers to fill our GPU memory. Then, we repeatedly transfer layers from CPU to GPU memory, perform forward passes through the layers, and then discard them from GPU memory until we have completed the number of forward passes corresponding to the number of layers in LLaMA-65B. To compare with MCNC, we first create MCNC parameters corresponding to 1% of the full LLaMA-65B. Then, we follow a similar protocol of generating weights until GPU memory is full, performing forward passes through the generated layers, and discarding the layers from GPU memory. As the timing of MCNC generation depends on the generator size, we perform this experiment with an MLP with three hidden layers and different widths of the hidden layer. We conduct these experiments using a single Nvidia RTX A6000 GPU with 48GB of memory. Our results demonstrate that, despite the computational overhead of generating parameters, MCNC with an appropriate generator size can achieve a significant speedup during inference, in addition to its primary contribution of model compression.
> >
> > |          | Latency (s) |
> > | -------- | -------------------------------------- |
> > | Baseline |                  14.7                      |
> > | MCNC (hidden size=64) |            5.5                |
> > | MCNC (hidden size=128) |            5.8               |
> > | MCNC (hidden size=256) |             7.8              |
> > | MCNC (hidden size=512) |              11.1           |
> > | MCNC (hidden size=1024) |             20.4              |
> >
> > In addition, as CPU-GPU communication is shared between GPUs, we believe our method has advantages in multi-GPU deployments. To test this, we performed the above test across four GPUs (A6000) simultaneously and reported the average latency below. Given that the CPU-GPU communication is the main bottleneck here, this results in an increase in latency for the baseline, whereas the inference latency with MCNC remains primarily unchanged.
> >
> > |          | Latency (s) |
> > | -------- | -------------------------------------- |
> > | Baseline |                  22.2                      |
> > | MCNC (hidden size=64) |            5.6                |
> > | MCNC (hidden size=128) |            6.0               |
> > | MCNC (hidden size=256) |             7.6              |
> > | MCNC (hidden size=512) |              11.6           |
> > | MCNC (hidden size=1024) |             21.7              |
> >
> > Lastly, we thank the reviewer for highlighting the advantages of achieving extreme compression rates. We reemphasize that this setting is particularly valuable in scenarios such as 1) deploying models on resource-constrained devices with limited memory, 2) storing many expert networks in GPU memory to enable efficient on-demand expert model inference, and 3) model communication in low-bandwidth environments.

---

> > > ### Comment · Reviewer_97Hi · 2024-12-02
> > >
> > > Thanks for showing the latency numbers. I appreciate the authors' effort to show computation performance numbers in a simple setup. I remain positive in the work.

---

### Author Response · Authors · 2024-11-25
**Engaging in Final Discussions and Requesting Reevaluations as the Discussion Period Closes**

Dear Reviewers,

Thank you once again for your time and dedication to maintaining the high standards of the ICLR conference. As we near the end of the discussion period, we would like to check in and engage in constructive dialogue to address any remaining concerns you may have.

We have provided detailed responses to your thoughtful feedback and would be happy to engage with you further to clarify or expand on any points. Please let us know if there are additional questions we can address.

Best,
The Authors

---

### Meta-Review · Area_Chair_Qa4R · 2024-12-24

**Metareview:**

The key idea behind this submission was remarked on positively by multiple reviewers, with the idea being considered interesting and 'very simple yet powerful'. Its deep roots in the random projections/random hashing literature were remarked on positively, highlighting the potential for inspiring future work. The evaluation was considered thorough, with extensive ablation studies provided and some demonstration of real-world speedups presented.

Some remaining criticisms post-rebuttal are the missing theoretical underpinnings / methodological motivations several authors asked about. Doubt also remains in one reviewer's mind about the expressivity of the method.

While it would have been strongly beneficial for one strong proponent of the work to emerge among the reviewers, given the generally positive reception of this work, I am happy to recommend this submission for acceptance.

**Additional Comments On Reviewer Discussion:**

Good discussion, resulting in a favourable remark by reviewer 97Hi and a score increase by reviewer dj4q.

---

### Decision · Program_Chairs · 2025-01-22

Accept (Poster)